# Reversible Residual Normalization Alleviates Spatio-Temporal Distribution Shift

## Abstract

Distribution shift severely degrades the performance of deep forecasting models. While this issue is well-studied for individual time series, it remains a significant challenge in the spatio-temporal domain. Effective solutions like instance normalization and its variants can mitigate temporal shifts by standardizing statistics. However, distribution shift on a graph is far more complex, involving not only the drift of individual node series but also heterogeneity across the spatial network where different nodes exhibit distinct statistical properties. To tackle this problem, we propose Reversible Residual Normalization (RRN), a novel framework that performs spatially-aware invertible transformations to address distribution shift in both spatial and temporal dimensions. Our approach integrates graph convolutional operations within invertible residual blocks, enabling adaptive normalization that respects the underlying graph structure while maintaining reversibility. By combining Center Normalization with spectral-constrained graph neural networks, our method captures and normalizes complex Spatio-Temporal relationships in a data-driven manner. The bidirectional nature of our framework allows models to learn in a normalized latent space and recover original distributional properties through inverse transformation, offering a robust and model-agnostic solution for forecasting on dynamic spatio-temporal systems.

## 1 Introduction

Spatio-temporal forecasting plays a critical role in understanding and predicting dynamic systems across diverse domains, from urban traffic management (Guo et al., 2019; Shi et al., 2020; Fang et al., 2019) and environmental monitoring (Liang et al., 2023; Hu et al., 2022; Jin et al., 2023) to public health surveillance and resource allocation. These applications require models that can accurately capture both the spatial dependencies among interconnected locations and the temporal dynamics that govern system evolution. While recent advances in deep learning have significantly improved forecasting capabilities through sophisticated architectures combining graph neural networks with temporal modeling components, a fundamental challenge remains largely unaddressed: the presence of distribution shift in spatio-temporal data.

Distribution shift manifests when the statistical properties of data change over time or vary across spatial locations, violating the standard assumption that training and testing data follow identical distributions. In spatio-temporal systems, this challenge is particularly acute due to its dual nature. Temporally, the statistical characteristics of observations at any given location may drift over time due to seasonal variations, policy changes, or evolving system dynamics. Spatially, different locations often exhibit distinct statistical properties influenced by local conditions, geographic features, or functional characteristics, even when they share underlying correlational patterns. This Spatio-Temporal distribution shift (Hu et al., 2023) can severely degrade model performance, as networks trained on historical data struggle to generalize when confronted with evolved temporal patterns or diverse spatial characteristics during inference.

Existing approaches to handling distribution shift have primarily focused on univariate or multivariate time series forecasting, where normalization techniques have proven effective. Methods such as instance normalization (Ulyanov et al., 2016) and its variants aim to standardize temporal sequences by removing non-stationary components through statistical transformations. However, these techniques were designed

for temporal data and do not naturally extend to the graph-structured nature of spatio-temporal systems. Simply applying temporal normalization independently to each node fails to address the spatial dimension of the shift problem and may even disrupt important spatial relationships encoded in the graph structure. Most spatio-temporal forecasting methods concentrate on designing sophisticated architectures to capture correlations but implicitly assume stationarity or homogeneity in the data distribution, leaving a critical gap between model assumptions and real-world data characteristics.

To bridge this gap, we propose a novel framework that extends normalization principles to the spatio-temporal domain through carefully designed invertible transformations. We introduce **Reversible Residual Normalization** (RRN), which combines graph-aware transformations with invertible residual architectures to address distribution shift in both spatial and temporal dimensions. Our method consists of three key components: Center Normalization (Qi et al., 2023), a Lipschitz-continuous alternative (Zha et al., 2021; Behrmann et al., 2019; Park et al., 2024) that maintains invertibility while removing distributional shifts; graph convolutional operations within invertible residual blocks (Park et al., 2024) for spatially-aware normalization; and spectral normalization (Miyato et al., 2018) to ensure the entire transformation remains invertible. The bidirectional nature of our framework is essential for practical forecasting. The forward transformation maps diverse spatio-temporal data into a normalized latent space where patterns become more regular and easier to learn. After a base forecasting model generates predictions in this latent space, the inverse transformation restores location-specific and time-varying characteristics. This symmetric design allows our framework to serve as a model-agnostic wrapper that can enhance any existing spatio-temporal forecasting architecture. Our contributions can be summarized as follows. We identify and formalize the problem of Spatio-Temporal distribution shift in graph-based forecasting systems, highlighting its dual nature and the limitations of existing normalization approaches. Finally, through extensive experiments on real-world benchmarks, we show that our method consistently improves multiple forecasting models, validating the importance of addressing distribution shift in spatio-temporal systems.

## 2 Related works

### 2.1 Distribution Shift in Time Series and Spatio-Temporal Data

Distribution shift arises from non-stationarity, causing significant discrepancies between training and testing distributions Kim et al. (2021); Liu et al. (2023b); Fan et al. (2023; 2025). To mitigate this, instance normalization and its variants, such as reversible and adaptive normalization effectively standardize temporal sequences by removing non-stationary components, yet they typically process nodes independently. Other advanced approaches explicitly model distribution shifts by learning intra-space and inter-space variations, but these remain confined to the temporal dimension. Consequently, the spatial dimension of distribution shift is largely neglected, applying temporal normalization strategies directly to spatio-temporal data ignores the underlying graph structure, failing to account for spatial heterogeneity where distinct nodes exhibit unique statistical properties.

### 2.2 Invertible Networks

Invertible neural networks learn bijective transformations that allow for exact computation of both forward and inverse mappings, facilitating tasks such as generative modeling and density estimation (Dinh et al., 2017; Zhai et al., 2025; Liu et al., 2019; Behrmann et al., 2019; Kingma & Dhariwal, 2018). While normalizing flows typically rely on coupling layers to construct invertible mappings (Dinh et al., 2017; Kingma & Dhariwal, 2018; Dinh et al., 2015; Kingma et al., 2016), Invertible Residual Networks (Behrmann et al., 2019; Zha et al., 2021; Park et al., 2024) extend this capability to standard residual architectures. Rather than relying on partitioning dimensions, these models utilize residual blocks (He et al., 2016) that guarantee invertibility provided the residual function satisfies specific Lipschitz continuity conditions (requiring a constant strictly less than one). This constraint is often enforced via spectral normalization to enable exact inverse computation through fixed-point iteration. Despite their success in time series, the potential of invertible architectures to simultaneously address spatial and temporal distribution shift while preserving graph topology remains underexplored.

## 3 Problem Formulations

### 3.1 Spatio-Temporal Forecasting

Let a spatio-temporal system be represented by a graph $\mathcal{G} = (V, E)$, where $V$ is a set of $N$ spatial locations (e.g., sensors, regions) with $|V| = N$, and $E$ is a set of edges representing the spatial relationships between these locations. At each discrete time step $t$, the system exhibits a feature matrix $X^{(t)} \in \mathbb{R}^{N \times D}$, where $D$ is the number of features recorded at each location. This feature matrix is also referred to as a graph signal.

The task of spatio-temporal forecasting aims to predict future system states based on historical observations. Given a lookback window of length $L$, the historical data can be denoted as a tensor $\mathcal{X}_{t-L+1:t} = (X^{(t-L+1)}, \ldots, X^{(t)}) \in \mathbb{R}^{L \times N \times D}$. The objective is to forecast the subsequent data over a horizon window of length $H$, denoted as $\mathcal{Y}_{t+1:t+H} = (X^{(t+1)}, \ldots, X^{(t+H)}) \in \mathbb{R}^{H \times N \times D}$.

Formally, the goal is to learn a function $f_\theta$ parameterized by $\theta$ that maps the historical observations and the graph structure to the future sequence:

$$\hat{\mathcal{Y}}_{t+1:t+H} = f_\theta(\mathcal{X}_{t-L+1:t}; \mathcal{G})$$

where $\hat{\mathcal{Y}}_{t+1:t+H}$ represents the predicted future sequence.

### 3.2 Distribution Shift in Spatio-Temporal Series

A primary challenge in real-world spatio-temporal forecasting is the non-stationarity of the data, which manifests as a distribution shift. Many deep learning models implicitly assume that the data-generating process is consistent between the training and test periods, an assumption that is frequently violated in practice. This discrepancy occurs as the joint distribution of the observed spatio-temporal data changes over time, hindering a model's ability to generalize from past observations to future predictions.

Formally, for any two distinct time steps $t_u$ and $t_v$, the underlying conditional probability that governs the system's evolution is not constant. This distribution shift can be expressed as:

$$P(\mathcal{Y}_{t_u+1:t_u+H} | \mathcal{X}_{t_u-L+1:t_u}; \mathcal{G}) \neq P(\mathcal{Y}_{t_v+1:t_v+H} | \mathcal{X}_{t_v-L+1:t_v}; \mathcal{G})$$

Therefore, our objective is not only to predict future spatio-temporal states but also to explicitly model and mitigate these distribution shifts to enhance forecasting accuracy and generalization. To address this, we propose extending the principles of normalizing flows, which have shown promise in handling temporal distribution shifts, to the more complex spatio-temporal domain.

## 4 Methodology

### 4.1 Invertible Residual Structure

Residual structures (He et al., 2016) have become a fundamental building block in modern deep learning architectures. A residual block takes the form:

$$H(x) = x + G(x) \tag{1}$$

Without loss of generality, we define $x \in \mathbb{R}^d$ is the input, $G : \mathbb{R}^d \to \mathbb{R}^d$ represents a learnable transformation, and the output $H(x)$ combines the identity mapping with the learned residual $G(x)$. This formulation facilitates gradient flow during training and enables the construction of very deep networks.

For our purposes, we are particularly interested in when such residual structures are invertible, meaning there exists a unique inverse mapping $H^{-1}$ such that $H^{-1}(H(x)) = x$ for all $x$ in the domain. Invertibility is desirable in our context because it allows us to transform data distributions bidirectionally, which is essential for modeling distribution shifts in spatio-temporal forecasting.

A sufficient condition for the invertibility of a residual block is provided by the following lemma, which constrains the behavior of the residual function $G$:

**Lemma 1** (Sufficient Condition for Invertible Residual Blocks (Behrmann et al., 2019; Park et al., 2024)). *Let $H(x) = x + G(x)$ be a residual block, where $G : \mathbb{R}^d \to \mathbb{R}^d$. If the Lipschitz constant of $G$ satisfies:*

$$L(G) = \sup_{x_1 \neq x_2} \frac{\|G(x_1) - G(x_2)\|}{\|x_1 - x_2\|} < 1 \tag{2}$$

*then $H$ is invertible.*

The intuition behind this lemma is that when $G$ is a contraction mapping (i.e., $L(G) < 1$), the residual block $H$ becomes a perturbation of the identity function that preserves bijectivity. Under this condition, the inverse $x = H^{-1}(z)$ for a given output $z$ can be computed through fixed-point iteration as shown in Algorithm 1.

---

**Algorithm 1:** Inverse of Residual Block via Fixed-Point Iteration

---

**Input:** Output $\mathbf{x}^{(\ell)}$ from residual block, residual function $G$, number of iterations $N$
**Output:** input of residual block $\mathbf{x}^{(\ell-1)}$
**1** $\mathbf{x} \leftarrow \mathbf{x}^{(\ell)}$;
**2 for** $m = 1, \dots, N$ **do**
**3** $\quad \mathbf{x} \leftarrow \mathbf{x} - G(\mathbf{x})$;
**4 return x**

---

In practice, enforcing the Lipschitz constraint $L(G) < 1$ requires careful design of the residual function $G$. When $G$ is composed of linear transformations (such as convolutions) followed by nonlinear activations, a common approach is to constrain the spectral norm of each linear layer. Specifically, if $G$ consists of multiple layers with weight matrices $W_1, W_2, \dots, W_i$, we can enforce:

$$\tilde{W}_i = \begin{cases} cW_i/\tilde{\sigma}_i, & \text{if } c/\tilde{\sigma}_i < 1 \\ W_i, & \text{else} \end{cases} \tag{3}$$

where $\tilde{\sigma}_i = \|W_i\|_2$ denotes the spectral norm (largest singular value) of $W_i$, and $c < 1$ are chosen such that the overall Lipschitz constant of $G$ remains below 1. The spectral norm can be efficiently approximated using power iteration methods (Miyato et al., 2018; Gouk et al., 2021) during training. This framework of invertible residual blocks provides the foundation for our approach to handling distribution shifts in spatio-temporal forecasting, as we will elaborate in the following sections.

### 4.2 Center Normalization instead of Instance Normalization

Instance Normalization (IN) (Ulyanov et al., 2016; Fan et al., 2025; Liu et al., 2023b; Kim et al., 2021) has demonstrated significant effectiveness in mitigating distribution shift across various domains. By normalizing each sample independently, IN reduces dependence on specific distributional statistics, thereby improving generalization. However, Instance Normalization is not Lipschitz continuous, which contradicts the invertibility requirement in Lemma 1.

For input $\mathcal{X}_{t-T+1:t} \in \mathbb{R}^{T \times N \times D}$ representing temporal features at a spatial location, we omit the time subscripts for brevity and denote it as $\mathcal{X}$. Instance

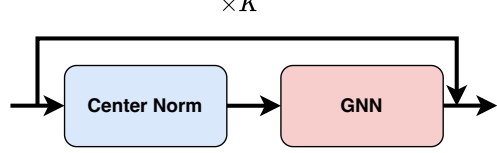

Figure 1: Architecture of the Invertible Residual Block. The residual structure consists of two main components: Center Normalization (CN) removes temporal mean shifts while maintaining Lipschitz continuity, followed by a Graph Neural Network (GNN) layer with spectral normalization to capture spatial dependencies.

Normalization is defined as:

$$\text{IN}(\mathcal{X}) = \gamma \odot z + \beta,$$
$$\text{where} \quad z = \frac{y}{\text{Std}(y)}, \quad y = \left(I - \frac{1}{T}\mathbf{1}\mathbf{1}^\top\right)\mathcal{X}. \tag{4}$$

The Jacobian matrix of IN reveals its discontinuity:

$$J_z(\mathcal{X}) = \frac{\partial z}{\partial \mathcal{X}} = \frac{1}{\text{Std}(y)}\left(I - \frac{1}{T}\mathbf{1}\mathbf{1}^\top\right)\left(I - \frac{yy^\top}{\|y\|_2^2}\right). \tag{5}$$

When $\text{Std}(y)$ approaches 0, the Jacobian entries approach $\infty$, violating Lipschitz continuity. This causes training instability and precludes invertibility. To address this, we introduce **Center Normalization**(Qi et al., 2023):

$$\text{CN}(\mathcal{X}) = \gamma \odot \alpha\left(I - \frac{1}{T}\mathbf{1}\mathbf{1}^\top\right)\mathcal{X} + \beta, \tag{6}$$

where $\alpha$ is a controllable scaling parameter, $\gamma$ and $\beta \in \mathbb{R}^{N \times D}$ are learnable parameters. CN removes the mean without variance normalization, ensuring Lipschitz continuity. The Jacobian of the Center Normalization is $\frac{\partial \text{CN}(\mathcal{X})}{\partial \mathcal{X}} = \alpha\left(I - \frac{1}{T}\mathbf{1}\mathbf{1}^\top\right)$. When $\gamma = \mathbf{1}$ and $\beta = \mathbf{0}$, Center Normalization has Lipschitz constant $\text{Lip}(\text{CN}_\mathcal{X}) = \alpha$, it's trivial to verify $\|\text{CN}(\mathcal{X}_1) - \text{CN}(\mathcal{X}_2)\| \le \alpha\|\mathcal{X}_1 - \mathcal{X}_2\|$. By setting $\alpha \le 1$, CN satisfies the invertibility condition in Lemma 1. The centering operation preserves distribution shift mitigation by removing mean shifts, a common form of temporal distribution shift.

### 4.3 Overall Structure

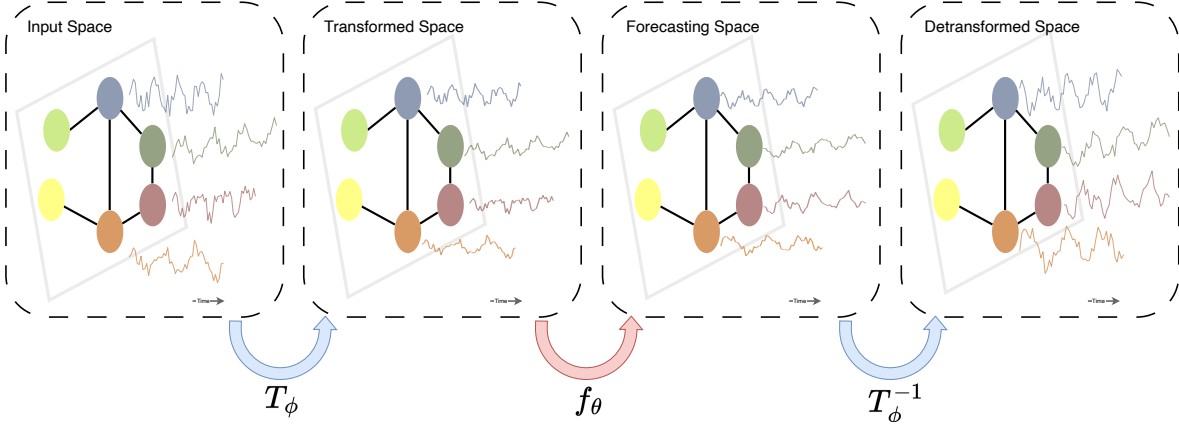

Figure 2: Overview of the Reversible Residual Normalization framework for spatio-temporal forecasting. The framework operates through four stages: (1) **Input Space** contains raw spatio-temporal data with distribution shifts across nodes and time; (2) **Transformed Space** where the invertible transformation $T_\phi$ normalizes the data by removing Spatio-Temporal distribution shifts while preserving correlational structures; (3) **Forecasting Space** where any forecasting model $f_\theta$ makes predictions in the stationary latent space; (4) **Detransformed Space** where the inverse transformation $T_\phi^{-1}$ restores the original distributional properties to produce final predictions.

After applying Center Normalization, we incorporate a graph convolution module within the invertible residual block to capture spatial dependencies. Following the framework in Section 3.1, we introduce a Lipschitz-constrained Graph Convolutional Network (GCN) layer (Kipf & Welling, 2017; Park et al., 2024) to ensure the overall residual block satisfies the invertibility condition.

### 4.3.1 Spectral Normalization for GCN

A standard GCN layer with residual connection can be formulated as:

$$H(X^{(t)}) = X^{(t)} + \sigma(\hat{A}X^{(t)}W) = X^{(t)} + g(X^{(t)}), \tag{7}$$

where $X^{(t)} \in \mathbb{R}^{N \times D}$ is the input node representation, we omit the time superscripts for brevity and denote it as $X$, $\hat{A} = \tilde{D}^{-\frac{1}{2}}\tilde{A}\tilde{D}^{-\frac{1}{2}}$ is the normalized adjacency matrix with $\tilde{A} = A + I$ (adjacency matrix with self-loops) and $\tilde{D}$ is the diagonal degree matrix, $W \in \mathbb{R}^{d \times d}$ is the learnable weight matrix, and $\sigma(\cdot)$ is a Lipschitz continuous activation function (e.g., ReLU, tanh). According to Lemma 1, the residual block in Eq. equation 7 is invertible if the Lipschitz constant of the residual function $g(X) = \sigma(\hat{A}XW)$ satisfies:

$$\text{Lip}(g) = \sup_{X_1 \neq X_2} \frac{\|g(X_1) - g(X_2)\|_2}{\|X_1 - X_2\|_2} < 1. \tag{8}$$

For contractive activation functions like ReLU and tanh where $\text{Lip}(\sigma) < 1$, the condition is satisfied if:

$$\sup_{X \neq 0} \frac{\|\hat{A}XW\|_2}{\|X\|_2} < 1. \tag{9}$$

This supremum is upper bounded by:

$$\sup_{X \neq 0} \frac{\|\hat{A}XW\|_2}{\|X\|_2} \leq \|\hat{A}\|_2 \|W\|_F, \tag{10}$$

where $\|\cdot\|_2$ denotes the spectral norm (largest singular value), and $\|W\|_F$ is the Frobenius norm of $W$, while strict invertibility requires bounding the spectral norm $\|W\|_2$, we strictly constrain the Frobenius norm $\|W\|_F$ in practice as an upper bound ($\|W\|_2 \leq \|W\|_F$) for computational efficiency, same operation as (Park et al., 2024), bypass the inconvenience of power iteration. For the normalized adjacency matrix $\hat{A}$ in GCN, we have $\|\hat{A}\|_2 = 1$. Therefore, to ensure $\text{Lip}(g) < c$ for some $c < 1$, we normalize the weight matrix $W$ according to Eq. equation 3 after each gradient descent step. This spectral normalization ensures the invertibility of the residual block throughout training.

### 4.3.2 Reversible Residual Normalization

We now combine Center Normalization and the Lipschitz-constrained GCN into an invertible residual block. The complete block is defined as:

$$H(\mathcal{X}^{(\ell)}_{t-T+1:t}) = \mathcal{X}^{(\ell)}_{t-T+1:t} + \sigma(\hat{A} \cdot \text{CN}(\mathcal{X}^{(\ell)}_{t-T+1:t}) \cdot W), \tag{11}$$

where $\text{CN}(\cdot)$ is Center Normalization from Eq. equation 6. The Lipschitz constant of this block by chain rules of Lipschitz satisfies:

$$\text{Lip}(g) \leq \text{Lip}(\sigma) \cdot \|\hat{A}\|_2 \cdot \text{Lip}(\text{CN}) \cdot \|W\|_F. \tag{12}$$

## 4.4 Bidirectional Transformation Framework

Our approach transforms spatio-temporal data into a stationary latent space, performs forecasting, and transforms predictions back to the original space. Formally, let $\mathcal{X}_{t-L+1:t}$ denote the historical observations. We define:

**Forward Transformation (to stationary space):**

$$\mathcal{Z} = T_\phi(\mathcal{X}_{t-L+1:t}; \mathcal{G}), \tag{13}$$

where $H^{(\ell)}$ denotes the $\ell$-th invertible spatio-temporal block from Eq. equation 11.

**Prediction in stationary space:**

$$\hat{\mathcal{Z}}_{t+1:t+H} = f_\theta(\mathcal{Z}; \mathcal{G}), \tag{14}$$

where $f_\theta$ is an arbitrary spatio-temporal forecasting model operating in the stationary latent space.

**Inverse Transformation (back to original space):**

$$\hat{\mathcal{Y}}_{t+1:t+H} = T_\phi^{-1}(\hat{\mathcal{Z}}_{t+1:t+H}; \mathcal{G}), \tag{15}$$

where each $(H^{(\ell)})^{-1}$ is computed via fixed-point iteration due to guaranteed invertibility.

The key insight is that by transforming to a stationary latent space $\mathcal{Z}$, we mitigate distribution shift: the conditional distribution $P(\hat{\mathcal{Z}}_{t+1:t+H}|\mathcal{Z}; \mathcal{G})$ becomes approximately invariant across time, enabling more robust forecasting with any backbone model $f_\theta$.

## 5 Experiments

Table 1: Performance comparison of baseline methods and their RRN-enhanced variants across different datasets and prediction horizons. Bold values indicate better performance (lower is better) in each baseline vs. RRN pair.

| Dataset | Horizon | GWavenet | | GWavenet + RRN | | DCRNN | | DCRNN + RRN | | GRUGCN | | GRUGCN + RRN | | AGCRN | | AGCRN + RRN | |
|---|---|---|---|---|---|---|---|---|---|---|---|---|---|---|---|---|---|
| | | MAE | RMSE | MAE | RMSE | MAE | RMSE | MAE | RMSE | MAE | RMSE | MAE | RMSE | MAE | RMSE | MAE | RMSE |
| LargeST-SD | 3 | 15.61 | 24.73 | **15.19** | **23.84** | 18.72 | **29.10** | **18.31** | 29.18 | 18.92 | 30.34 | **18.24** | **28.85** | 17.23 | 26.78 | **17.93** | **28.51** |
| | 6 | 19.72 | 30.86 | **19.48** | **30.37** | 26.11 | **40.35** | **25.46** | 40.67 | 27.66 | 43.27 | **27.03** | **41.01** | 21.38 | 32.76 | **21.41** | **34.18** |
| | 12 | 25.08 | **37.95** | **24.82** | 38.65 | 40.06 | 60.27 | **36.64** | **57.66** | 45.16 | 67.92 | **38.32** | **56.43** | **26.57** | **40.41** | 26.87 | 41.63 |
| SDWPF | 3 | 58.72 | **110.61** | **58.20** | 113.60 | 64.06 | 119.12 | **60.18** | **114.68** | 62.32 | 119.66 | **61.29** | **114.13** | 60.59 | 111.50 | **57.52** | **108.88** |
| | 6 | 83.34 | **151.75** | **83.10** | 156.56 | 90.13 | 161.25 | **85.65** | **158.15** | 87.78 | 161.84 | 88.85 | **157.06** | 85.54 | 151.76 | **82.21** | **150.25** |
| | 12 | 114.95 | 204.17 | **114.18** | **203.98** | 126.31 | 216.19 | **122.95** | **210.47** | 125.24 | 215.47 | **123.54** | **210.12** | 122.19 | 204.03 | **116.87** | **200.24** |
| MetrLA | 3 | 2.75 | 5.22 | **2.73** | **5.19** | 2.91 | 5.58 | **2.84** | **5.40** | 3.03 | 5.90 | **2.91** | **5.69** | 2.88 | 5.49 | **2.71** | **5.21** |
| | 6 | 3.16 | 6.27 | **3.16** | **6.24** | 3.40 | 6.74 | **3.30** | **6.51** | 3.70 | 7.38 | **3.34** | **7.02** | 3.27 | 6.51 | **3.09** | **6.17** |
| | 12 | 3.68 | 7.44 | **3.66** | **7.39** | 4.13 | 8.25 | **3.93** | **7.90** | 4.74 | 9.30 | **4.33** | **8.65** | 3.66 | 7.48 | **3.36** | **7.33** |

### 5.1 Experimental setup

**Datasets** Our evaluation encompasses three diverse spatio-temporal datasets with distinct characteristics and scales. The **METR-LA** dataset captures traffic speed patterns from 207 highway loop detectors across Los Angeles County, with measurements recorded at 5-minute intervals over four months from March to June 2012. This widely-used benchmark represents a medium-scale urban traffic network with well-established spatial connectivity. The **LargeST-SD** dataset (Liu et al., 2023a), a subset of the large-scale LargeST collection, provides traffic speed data from sensors distributed across San Diego with extended temporal coverage spanning multiple years at 5-minute sampling frequency, enabling the study of long-term distribution shifts and seasonal variations. The **SDWPF** (Wind Turbine Power) dataset (Zhou et al., 2024) provides wind power generation data from 134 turbines collected over 24 months, with each turbine recording 19 dynamic features including operational parameters from SCADA systems and meteorological variables from ERA5 reanalysis data, representing a distinct application domain with weather-driven dynamics and energy system constraints. For graph construction, METR-LA and LargeST-SD employ distance-based connectivity where edges are established between sensors using Gaussian kernel thresholding on geographic distances, while SDWPF utilizes spatial proximity based on turbine locations to capture wind pattern propagation across the turbine array. Each dataset is split chronologically into 70% training, 10% validation, and 20% testing sets to reflect realistic forecasting scenarios where models must generalize to future time periods.

**Baselines** We compare our model against several state-of-the-art baseline methods designed to address non-stationarity and distribution shift in time series forecasting. **RevIN** (Kim et al., 2021) is a model-agnostic method that normalizes each time series instance by its own mean and variance to mitigate distribution shift. It then uses these same statistics to symmetrically denormalize the model's output, restoring

Table 2: Performance comparison of DCRNN with different enhancement methods across datasets and prediction horizons. Bold values indicate better performance compared to baseline DCRNN (lower is better).

| Dataset | Horizon | DCRNN | | DCRNN + RevIN | | DCRNN + Dish-TS | | DCRNN + SAN | | DCRNN + RRN | |
|---|---|---|---|---|---|---|---|---|---|---|---|
| | | MAE | RMSE | MAE | RMSE | MAE | RMSE | MAE | RMSE | MAE | RMSE |
| LargeST-SD | 3 | 18.72 | **29.10** | 19.40 | 31.51 | 19.25 | 30.31 | 21.63 | 34.11 | **18.31** | 29.18 |
| | 6 | 26.11 | **40.35** | 29.22 | 46.22 | 28.03 | 43.21 | 32.53 | 49.78 | **25.46** | 40.67 |
| | 12 | 40.06 | 60.27 | 50.36 | 76.06 | 43.83 | 63.34 | 54.31 | 79.78 | **36.64** | **57.66** |
| SDWPF | 3 | 64.06 | 119.12 | 62.32 | 120.76 | 61.50 | 117.53 | 70.77 | 125.27 | **60.18** | **114.68** |
| | 6 | 90.13 | 161.25 | 88.91 | 164.25 | 88.11 | 158.89 | 96.88 | 164.92 | **85.65** | **158.15** |
| | 12 | 126.31 | 216.19 | 127.88 | 221.45 | 126.12 | 212.19 | 131.35 | 214.58 | **122.95** | **210.47** |
| MetrLA | 3 | 2.91 | 5.58 | 2.97 | 5.73 | 2.93 | 5.61 | 3.23 | 6.31 | **2.84** | **5.40** |
| | 6 | 3.40 | 6.74 | 3.57 | 7.11 | 3.43 | 6.79 | 3.90 | 7.75 | **3.30** | **6.51** |
| | 12 | 4.13 | 8.25 | 4.53 | 8.97 | 4.19 | 8.32 | 4.99 | 9.74 | **3.93** | **7.90** |

Table 3: Performance comparison across different prediction horizon. Bold values indicate better performance in each baseline vs. RRN pair (lower is better).

| Predict length Metric Avg. | 12 | | 24 | | 48 | | 96 | |
|---|---|---|---|---|---|---|---|---|
| | MAE | RMSE | MAE | RMSE | MAE | RMSE | MAE | RMSE |
| GRUGCN | 89.41 | 163.96 | 121.84 | 214.71 | **168.38** | 279.15 | **219.12** | 334.28 |
| GRUGCN + RRN | **86.15** | **160.46** | **118.62** | **209.22** | 169.89 | **273.48** | 223.61 | **345.99** |
| DCRNN | 87.86 | 164.75 | 122.97 | 212.61 | **164.16** | 276.27 | **200.91** | 314.77 |
| DCRNN + RRN | **85.73** | **160.83** | **121.57** | **210.12** | 168.98 | **269.62** | 204.23 | **311.80** |

the original scale of the series. **Dish-TS** (Fan et al., 2023) This paradigm defines distribution shift as both intra-space (within inputs) and inter-space (between input and output) shifts. It uses a Dual-CONET framework to learn separate distribution coefficients for the input and output spaces, explicitly modeling the gap between them. **SAN** (Liu et al., 2023b) provides a more fine-grained approach, performing normalization on sub-series "slices" rather than on the entire instance to handle non-stationarity. The framework also features an independent module to predict the statistics of future slices for a more adaptive denormalization.

**Backbone models** Our proposed framework is model-agnostic and can be applied to arbitrary spatio-temporal forecasting models. To demonstrate its effectiveness and versatility, we select several mainstream backbone models with diverse architectures for evaluation. **DCRNN** (Li et al., 2018) Models traffic as diffusion process on directed graphs, combining graph diffusion convolutions with RNNs; **Graph WaveNet** (Wu et al., 2019) Combines graph convolutions with dilated causal convolutions and adaptive adjacency matrix for hidden spatial dependencies; **GRUGCN** (Gao & Ribeiro, 2021) Adopts a time-then-graph framework where GRU layers first encode temporal node/edge evolution, followed by GCN layers to capture spatial dependencies on the resulting static graph; **AGCRN** (Bai et al., 2020) Enhances GCN with Node Adaptive Parameter Learning (NAPL) to capture node-specific patterns and Data Adaptive Graph Generation (DAGG) to automatically infer spatial dependencies without requiring pre-defined graphs.

**Implementation Details** We maintain consistency with original implementations by using identical hyperparameters and configurations for all backbone models and baseline methods as reported in their respec-

tive papers. For our RRN framework, we employ two invertible residual blocks stacked sequentially. The controllable scaling parameter $\alpha$ in Center Normalization is set to 0.9, and we apply spectral normalization to each GCN weight matrix to enforce a Lipschitz constant of 0.9, ensuring invertibility according to Lemma 1. The hidden size of GCN layers within each residual block is set to 32. Across all experiments, we use the Adam optimizer with a learning rate of 5e-3, training for a maximum of 100 epochs with early stopping based on validation performance. The batch size is fixed at 64 for all datasets. All experiments are conducted on NVIDIA L40S GPUs. To ensure statistical reliability, we run each experiment with five different random seeds and report the averaged results along with their corresponding metrics.

## 5.2 Overall Performance

Tables 1 and 2 demonstrate the effectiveness of our Reversible Residual Normalization (RRN) framework. Table 1 shows consistent improvements when RRN is used with various baseline models. Notably, on LargeST-SD, DCRNN+RRN reduces MAE by 2.2%, 2.5%, and 8.5% for horizons 3, 6, and 12, respectively. The larger improvement for longer horizons suggests RRN becomes more valuable as distribution drift accumulates over time. The framework works well across different model types, including diffusion (DCRNN), recurrent (GRUGCN), and adaptive graph models (GWavenet, AGCRN). On SDWPF, AGCRN+RRN reduces MAE by 5.1%, 3.9%, and 4.4%, showing it effectively handles the strong distribution shifts in wind power data. Table 2 compares RRN against other normalization strategies on DCRNN. RRN consistently outperforms all baselines. Most notably, it achieves 27.3% lower MAE than RevIN on LargeST-SD for 12-step prediction. This gap highlights a key limit of temporal-only methods like RevIN, Dish-TS, and SAN. These methods ignore graph structure and treat nodes independently, which can disrupt spatial patterns and lower performance. In contrast, RRN uses graph convolutions within its invertible blocks to respect the spatial structure. Even compared to Dish-TS, which models distribution shifts explicitly, RRN maintains a 1.1% to 2.5% advantage on SDWPF. Overall, RRN delivers improvements ranging from 2.2% to 27.3%, proving that jointly modeling spatial and temporal dimensions is essential for accurate forecasting.

## 5.3 Performance On Different Prediction Lengths

Table 3 shows the performance of RRN on the SDWPF dataset across horizons ranging from 12 to 96 steps. The results show our method is strong in short to medium-term forecasting but offers limited gains for longer horizons. For 12 and 24 steps, RRN reduces MAE for GRUGCN (by 3.6% and 2.6%) and DCRNN (by 2.4% and 1.1%). This indicates RRN effectively reduces distribution shift when temporal drift is moderate. However, improvements decrease for 48 and 96 steps. This trend likely comes from the limit of graph-based models in capturing long-range dependencies, where performance drops naturally regardless of the normalization used. Long-term forecasting remains a challenge for spatio-temporal models compared to standard time series because the data scales with the number of nodes ($N$). The high computational cost and complexity make it difficult to extend these models to long sequences, making short to medium-term forecasting the most practical use case.

## 5.4 Extra Experiments

We conducted additional experiments to analyze the impact of model depth and the contribution of individual components. Figure 3 presents the results of these ablation studies.

**Impact of Residual Blocks.** We investigated how the number of invertible residual blocks affects forecasting performance. As illustrated in Figure 3a, increasing the number of stacked blocks does not yield significant improvements in prediction accuracy (MAE and RMSE remain stable). However, adding more blocks introduces substantial computational overhead, leading to slower training and inference times. Consequently, we adopt a configuration with fewer blocks to achieve an optimal balance between computational efficiency and model performance.

**Component Analysis.** To validate the effectiveness of our key components, we compared the full model against variants without Center Normalization (w/o CN) and without the invertible GNN module (w/o iGNN). Figure 3b and Figure 3c visualize the performance on MetrLA and SDWPF datasets across different

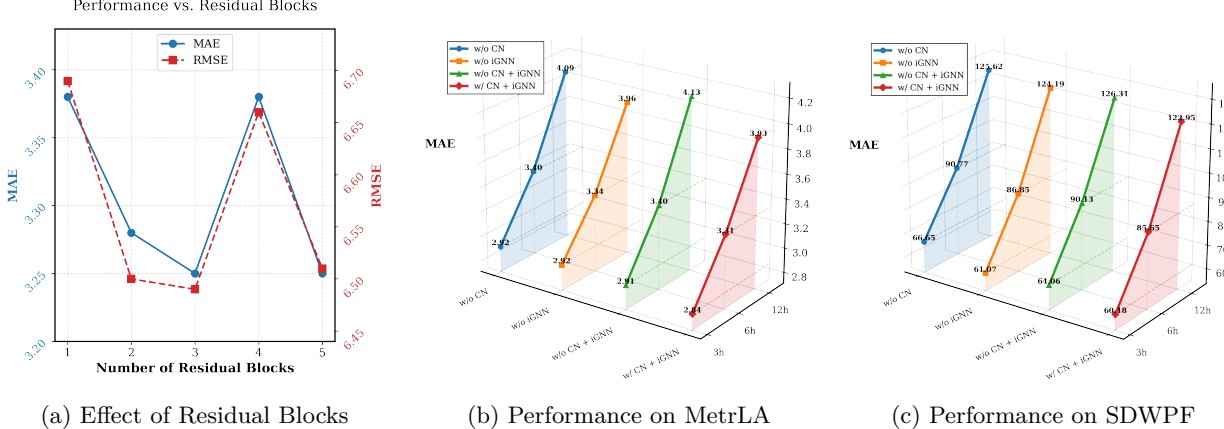

(a) Effect of Residual Blocks        (b) Performance on MetrLA        (c) Performance on SDWPF

Figure 3: Ablation studies and performance analysis. (a) The impact of the number of residual blocks on forecasting error. (b) and (c) 3D visualization of MAE across different prediction horizons and ablation variants on MetrLA and SDWPF datasets, respectively.

prediction horizons. The results demonstrate that the complete framework (w/ CN + iGNN) consistently achieves the lowest error rates. Removing either component leads to performance degradation, confirming that both the temporal normalization provided by CN and the spatial modeling from the iGNN are essential for effectively handling Spatio-Temporal distribution shifts.

# 6  Limitations

While the proposed framework demonstrates superior forecasting accuracy, two primary limitations exist regarding computational complexity and expressivity. First, the inverse transformation relies on fixed-point iteration, which requires $N$ sequential steps to recover the input. Although early termination based on convergence residuals can accelerate this process, the cumulative overhead across stacked blocks increases inference latency compared to standard instance normalization. Second, to ensure invertibility, we constrain the Lipschitz constant using the Frobenius norm of the weight matrices rather than the spectral norm (2-norm) typically estimated via power iteration (Behrmann et al., 2019). While substituting the spectral norm with the Frobenius norm eliminates the computational cost of singular value decomposition or iterative approximation (Park et al., 2024), the Frobenius norm serves as a conservative upper bound on the spectral norm. This stricter constraint unnecessarily restricts the feasible parameter space, which may limit the model's expressivity compared to methods that directly target the largest singular value.

# 7  Conclusion

This work addresses the challenge of distribution shift in spatio-temporal forecasting, where both temporal drift and spatial heterogeneity degrade model performance. We propose Reversible Residual Normalization, a framework that performs spatially-aware invertible transformations to normalize spatio-temporal data while preserving graph structure. Our approach combines Center Normalization, which maintains Lipschitz continuity unlike standard instance normalization, with graph convolutional operations within invertible residual blocks. Spectral normalization ensures the entire transformation remains invertible, enabling bidirectional mapping between original and normalized spaces. This model-agnostic framework can wrap any existing spatio-temporal forecasting architecture, allowing models to learn in a stationary latent space while recovering original distributional properties through inverse transformation. Our work demonstrates that explicitly accounting for both spatial and temporal dimensions of distribution shift through principled invertible transformations is essential for robust forecasting in dynamic spatio-temporal systems.

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
