# OpenReview forum: "Reversible Residual Normalization Alleviates Spatio-Temporal Distribution Shift"
_TMLR — Rejected by TMLR_

### Review · Reviewer_PVjn · 2025-12-08

**Summary Of Contributions:**

The paper proposes a Reversible Residual Normalization, an invertible residual block which is theoretically guaranteed with central normalization on graph convolutional networks. RRN is applied to the graph convolutional model to (1) encode the input time series into a latent space, (2) forecast a future latent space, (3) inverse-transform the future latent space back to the input space. Intensive experiments show the universal applicability of RRN to different architectures, algorithms, and datasets.

**Audience:**

Yes

**Audience Explanation:**

Spatio-temporal distribution shift problem is important in time-series forecasting, and the paper proposes a key insight that forecasting in latent space and inverse transforming into original sample space provides a better forecasting performance.

**Broader Impact Concerns:**

I don't have any ethical implication concerns.

**Claims And Evidence:**

No

**Claims Explanation:**

1. The paper clearly prove the reversible manner of central normalization.

1. However, the paper fails to convince how RRN can lead to better performance on the distribution shift problem.

1. I understand that it intuitively makes sense, but there is no explanation why "forecasting in latent space and inverse transforming into original sample space" can provide a better result.

**Requested Changes:**

1. Please clearly explain why forecasting in latent space -> inverse transform to sample space can lead to a better result.

1. Please clarify how RRN mitigates the spatio-temporal distribution shift problem. Is the center normalization mitigating the problem? Without considering the revertability, normalization itself is not a surprising component.

---

### Review · Reviewer_ooAa · 2025-12-11

**Summary Of Contributions:**

This paper tackles distribution shift in spatio-temporal forecasting, where performance often degrades because the data drift is not only temporal but also spatially heterogeneous across graph nodes. The authors propose Reversible Residual Normalization (RRN): a model-agnostic framework that uses invertible residual blocks incorporating graph operations to perform spatially-aware reversible transformations. The idea is to map inputs into a more stable, normalized latent space for training and then use the inverse transform to recover outputs back to the original distribution. The method also highlights combining center normalization with spectral-constrained GNN components to better respect graph structure while normalizing complex spatio-temporal relationships.

Strengths:

1.	It explicitly accounts for both temporal drift and cross-node spatial heterogeneity, which simpler normalization schemes often ignore.

2.	Training in a normalized space while being able to invert the transformation back to the original space is a clean and useful design for handling shifts without losing recoverability.

3.	Embedding graph operations (and using centering + spectral constraints) makes normalization structure-aware rather than treating each node/time series independently, better matching real spatio-temporal data.

Weakness:

1.	Similar normalization-based methods already exist in the spatio-temporal forecasting literature [1] and have achieved strong performance. However, the authors do not discuss the advantages of their approach over these works.

2.	All spatio-temporal forecasting backbones evaluated by the authors are from before 2022, making it difficult to judge whether the proposed method would also benefit more recent forecasting models.

3.	Experiments on some commonly used benchmark datasets are missing, such as PEMS04 and PEMS08.

4.	The paper lacks a clear summary and analysis of key hyperparameters, which seriously limits reproducibility.

5.	The proposed component increases computational complexity to some extent, but the performance gains are marginal. The paper also lacks an efficiency analysis to justify whether the improvements are worth the additional cost.

[1] Deng, Jinliang, et al. "St-norm: Spatial and temporal normalization for multi-variate time series forecasting." Proceedings of the 27th ACM SIGKDD conference on knowledge discovery & data mining. 2021.

**Audience:**

Yes

**Audience Explanation:**

Readers interested in machine learning and multivariate time-series modeling may find this work appealing.

**Claims And Evidence:**

No

**Claims Explanation:**

1.	The paper lacks theoretical analysis.

2.	The comparative experiments are insufficient, particularly due to the absence of comparisons with more specialized normalization baselines.

3.	There is no analysis of how much the proposed component improves the performance of more recent backbone models.

4.	The trade-off between efficiency and performance is not analyzed thoroughly.

5.	There is no hyperparameter analysis to ensure reproducibility.

**Requested Changes:**

1.	Compare against more specialized baselines, such as STNORM.

2.	Evaluate whether the proposed method can improve more recent models introduced in 2024–2025.

3.	Include additional datasets, such as PEMS04 and PEMS08.

4.	Provide a clearer motivation and a more explicit analysis of the technical contributions, e.g., clarifying advantages over STNORM.

5.	Provide a more comprehensive hyperparameter analysis.

6.	The trade-off between efficiency and performance should be analyzed.

7.	Since the performance gains are not significant, repeated runs and variance analysis are needed to show that the improvements are not due to chance.

---

### Review · Reviewer_raKs · 2025-12-14

**Summary Of Contributions:**

This paper focuses on spatiotemporal distribution shift in graph-based forecasting. They analyzed that the issues for the drifts are that each node can have time series drift over time and multiple nodes can have spatial heterogeneity also. Based on this, they propose their framework Reversible Residual Normalization (RRN). In their framework, they apply a graph aware invertible transformation to map original data into a stationary latent space. Then they do the forecasting based on the space. And Finally they apply the inverse transform to restore original predictions.

For the strengths: \
1: This paper gives a clear problem statement in the distribution shift, which are the temporal drift with spatial heterogeneity. They treat these problems together instead of treating them independently, which give us a clear way to understand their contribution. \
2: Their framework design gives a clear path: normalize first, then forecast, then denormalize and inverse transform. This is very clear and easy to understand.

For the weaknesses: \
1: Based on its method, its Reversible Residual Normalization's inverse mapping is computed through fixed point iteration. So it seems that for each block, it should recover the input through several sequential updates. Then if we have many stacked blocks, the inference latency may be aggregated to a huge overhead. This is a concerned issue since for the real-world deployment, we should always consider the latency of the model performance. \
2: Based on its method, we are not sure if Reversible Residual Normalization is robust enough to handle the scenario when variance changes are dominant. It seems that this framework uses center normalization to remove the mean shifts based on Lipschitz continuity. However, it didn't consider variance normalization. Variance changes may also a big concern during distribution shift since shift may also contains something like seasonal changes. More explanations and rigorous proofs are needed to clarify these points.

**Audience:**

Yes

**Audience Explanation:**

I think there are two points: \
1: This paper discuss a relevant issue in spatio-temporal distribution shift and proposes a "normalize first, then forecast, then denormalize and inverse transform" method to integrated with various backbones frameworks. This idea is an interesting topic in the field. \
2: Its invertible design and stability constraints mentioned in the paper is also an interesting topic to be read in the field.

**Broader Impact Concerns:**

Mentioned above. Nothing else. Thanks.

**Claims And Evidence:**

Yes

**Claims Explanation:**

I think it is partially supported. As I mentioned in the summary, its motivation and methodologies are very clear and the experiments show the evidence to support their design's performance. However, there are still some issues.

Besides the main weaknesses I mentioned above,
For the weaknesses: \
1: Based on its method, its Reversible Residual Normalization's inverse mapping is computed through fixed point iteration. So it seems that for each block, it should recover the input through several sequential updates. Then if we have many stacked blocks, the inference latency may be aggregated to a huge overhead. This is a concerned issue since for the real-world deployment, we should always consider the latency of the model performance. \
2: Based on its method, we are not sure if Reversible Residual Normalization is robust enough to handle the scenario when variance changes are dominant. It seems that this framework uses center normalization to remove the mean shifts based on Lipschitz continuity. However, it didn't consider variance normalization. Variance changes may also a big concern during distribution shift since shift may also contains something like seasonal changes. More explanations and rigorous proofs are needed to clarify these points.

Also, there are something else that needed to be improved: \
1: this paper also didn't give enough analysis and study on several things like accuracy-latency tradeoffs even if it has some existing issues. \
2: In performance comparison table 1, we do see its consistent improvements are not valid when comparing with AGCRN. Its LargeST-SD: AGCRN+RRN worse than AGCRN in some scenarios. More explanations should be provided.

**Requested Changes:**

Mainly as I mentioned above:

Besides the main weaknesses I mentioned above,
For the weaknesses: \
1: Based on its method, its Reversible Residual Normalization's inverse mapping is computed through fixed point iteration. So it seems that for each block, it should recover the input through several sequential updates. Then if we have many stacked blocks, the inference latency may be aggregated to a huge overhead. This is a concerned issue since for the real-world deployment, we should always consider the latency of the model performance. \
2: Based on its method, we are not sure if Reversible Residual Normalization is robust enough to handle the scenario when variance changes are dominant. It seems that this framework uses center normalization to remove the mean shifts based on Lipschitz continuity. However, it didn't consider variance normalization. Variance changes may also a big concern during distribution shift since shift may also contains something like seasonal changes. More explanations and rigorous proofs are needed to clarify these points.

Also, there are something else that needed to be improved: \
1: this paper also didn't give enough analysis and study on several things like accuracy-latency tradeoffs even if it has some existing issues. \
2: In performance comparison table 1, we do see its consistent improvements are not valid when comparing with AGCRN. Its LargeST-SD: AGCRN+RRN worse than AGCRN in some scenarios. More explanations should be provided.

---

### Decision · Action_Editor_kkm6 · 2026-01-23

**Recommendation:** Reject

**Additional Comments:**

As the Action Editor, I have carefully reviewed the original submission, the reviewers' comments, and the authors' rebuttal.

First, I would like to acknowledge the authors' proactive engagement during the discussion period. The timely and thorough responses to the reviewers' initial concerns were noted and appreciated. I also found the core idea of Reversible Residual Normalization to be conceptually interesting and the mathematical formulation to be coherent and well-motivated. The theoretical grounding of the framework is a strong point of the submission.

However, after a synthetic evaluation of the entire review process, I have determined that the Evidence criterion has not been met. While the authors provided additional analysis and clarifications in the rebuttal, they did not sufficiently resolve the fundamental concern.

Consequently, I am recommending a Reject decision. I encourage the authors to conduct a more rigorous empirical investigation, specifically focusing on isolating the effects of the proposed normalization from standard components and validating the method across a broader range of established datasets for a future submission.

**Audience:**

Yes

**Audience Explanation:**

The conceptual idea of combining reversible residual structures with normalization in the context of GNNs is of interest to a subset of the TMLR community working on graph representation learning and normalization techniques.

**Claims And Evidence:**

No

**Claims Explanation:**

While the proposed "Reversible Residual Normalization" (RRN) is theoretically interesting and the mathematical framework is well-constructed, the experimental evidence provided is insufficient to support the core claims of the paper.

Specifically, the following points remain unaddressed or inadequately supported even after considering the authors' rebuttal:

- As pointed out during the review process, the ablation studies do not convincingly demonstrate that the performance improvements stem from the proposed RRN mechanism. The results suggest that the gains may largely be attributed to Center Normalization (CN).
- The evaluation lacks validation on standard datasets in the field, such as the PEMS benchmarks. The omission of these standard comparisons, even if other datasets were used, makes it difficult to assess the actual utility and robustness of the method in a way that is convincing to the broader community.

Note on Comparison with Existing Methods: One reviewer pointed out that the improvement over similar established works (e.g., STNorm) is limited. While this observation has merit, I have determined that this falls outside the scope of the authors' specific claims. Since TMLR’s criteria focus on whether the claims are supported by evidence rather than on relative superiority or novelty, I have not used the limited improvement compared to existing methods as a reason for this rejection.